# Split-Ubiquitin Two-Hybrid Screen for Proteins Interacting with slToc159-1 and slToc159-2, Two Chloroplast Preprotein Import Receptors in Tomato (*Solanum lycopersicum*)

**DOI:** 10.3390/plants11212923

**Published:** 2022-10-30

**Authors:** Qi Wang, Jiang Yue, Chaozhong Zhang, Jianmin Yan

**Affiliations:** 1College of Agriculture, Guizhou University, Guiyang 550025, China; 2Vegetable Research Academy, Guizhou University, Guiyang 550025, China

**Keywords:** tomato, translocon of chloroplast outer membrane, Toc159, split-ubiquitin two-hybrid screen, interacting protein

## Abstract

The post-translational import of nuclear-encoded chloroplast preproteins is critical for chloroplast biogenesis, and the Toc159 family of proteins is the receptor for this process. Our previous work identified and analyzed the Toc GTPase in tomato; however, the tomato-specific transport substrate for Toc159 is still unknown, which limits the study of the function of the TOC receptor in tomato. In this study, we expand the number of preprotein substrates of slToc159 receptor family members using slToc159-1 and slToc159-2 as bait via a split-ubiquitin yeast two-hybrid membrane system. Forty-one specific substrates were identified in tomato for the first time. Using slToc159-1GM and slToc159-2GM as bait, we compared the affinity of the two bait proteins, with and without the A domain, to the precursor protein, which suggested that the A domain endowed the proproteins with subclass specificity. The presence of the A domain enhanced the interaction intensity of slToc159-1 with the photosynthetic preprotein but decreased the interaction intensity of slToc159-2 with the photosynthetic preprotein. Similarly, the presence of the A domain also altered the affinity of slToc159 to non-photosynthetic preproteins. Bimolecular fluorescence complementation (BiFC) analysis showed that A domain had the ability to recognize the preprotein, and the interaction occurred in the chloroplast. Further, the localization of the A domain in Arabidopsis protoplasts showed that the A domain did not contain chloroplast membrane targeting signals. Our data demonstrate the importance of a highly non-conserved A domain, which endows the slToc159 receptor with specificity for different protein types. However, the domain containing the information on targeting the chloroplast needs further study.

## 1. Introduction

In plant cells, chloroplasts are one of the many types of plastids that play crucial roles in photosynthesis and other metabolic pathways, including amino acid and lipid synthesis as well as nitrogen and sulfur assimilation [1,2]. According to the endosymbiotic origin theory, chloroplasts may have originated from prokaryotic cyanobacteria [3]. Chloroplast genetic material was transferred from the prokaryotic genome to the eukaryotic nucleus during evolution, and so today, most chloroplast proteins are encoded by the nucleus. Although chloroplasts have their own genomes, only about 100 chloroplast proteins are encoded by chloroplast genomes [4,5], while the remaining 95% (about 2000–2500 chloroplast proteins) are encoded by the nucleus. Consequently, plants have evolved mechanisms to coordinate the expression of nuclear genes with the developmental and functional state of the plastids [6,7]. To maintain the proper functioning of plants and their responses to developmental and external cues, the majority of nucleus-encoded chloroplast precursor proteins rely on the two multiprotein complexes, commonly known as the Toc/Tic (translocon at the outer/inner envelope membrane of chloroplasts), to translocate across the envelope [8].

Based on observations in *Pea*
*sativum* and *Arabidopsis*
*thaliana*, the components of the TOC complex include Toc159, Toc34, and Toc75. As core translocon components, Toc159 and Toc34 function as the initial receptors, and Toc75 forms the channel for protein translocation across the outer membrane [9,10]. In Arabidopsis, the Toc159 protein family is encoded by the Toc159, Toc132, Toc120, and Toc90 genes [11,12]. Chloroplast-targeted preproteins carry a cleavable N-terminal targeting sequence, called a transit peptide (TP), of 10–150 amino acids that contains all the information required for protein sorting from the cytoplasm to the chloroplast stroma [13]. TPs are recognized by two protein families of Toc GTPases: Toc159 and Toc34, which are exposed to the cytoplasm. In the targeting model, Toc159 was directly cross-linked with TP, demonstrating that Toc159 is the main preprotein receptor [9,14]. The binding of the TP to GTPase domains leads to changes in the dimerization of the receptor, GDP/GTP exchange, and GTP hydrolysis. Finally, the preproteins are inserted into the protein channel for the transmembrane in an ATP- and GTP-dependent manner [13]. Although the G domains of Toc159 and Toc34 are highly conserved, the former has an extra N-terminal sequence that is highly diverse and has intrinsically disordered acidic sequence domains [15]. Two previous studies have demonstrated that the GTPase domain mediates the targeting and insertion of Toc159 [16,17], although the field itself presumably does not contain any sorting signals, implicating that the A domain of Toc159 is of great significance for the specific binding of different types of preproteins [18].

However, almost all studies on preprotein transport to chloroplasts have thus far been based on observations in *Pisum sativum* and *Arabidopsis thaliana* [19,20,21,22]. Considering that Toc GTPase is highly conserved, we initially identified homologs of Toc GTPase in tomato [23]. In addition, tomato has distinct types of plastids, such as chloroplasts in the green leaves and chromoplasts in the ripe fruit, and thus, it is regarded as a new and interesting model plant for studying chloroplast protein transport. The availability of multiple plastid types has enabled us to elucidate the differences in preprotein imports between the chloroplast and other plastid types.

Our previous data showed that the expression levels of slToc159-1 were highest in green, photosynthetic tissues, whereas the expression levels of slToc159-2 were not significantly different between the photosynthetic and non-photosynthetic tissues, suggesting functional distinctions between the two slToc159 homologs. Additionally, based on previous studies [24,25,26], we believe that it is possible that such distinct Toc GTPases in tomato are also functionally distinct, each facilitating the import of a particular subset of preproteins. It is worth noting that, compared with the highly conserved G and M domains of Toc159 in other species [24], the sequence conservation of the A domain in the tomato Toc159 family is significantly weak, suggesting that the A domain may play an important role in the import of these specific proteins. However, the lack of reports on specific transport substrates for the tomato Toc GTPase limits this speculation.

Yeast two-hybrid (Y2H) technology is becoming increasingly popular for studying the interaction between proteins [27]. The Y2H system has been successfully used to analyze the chloroplast protein import apparatus [28,29,30]. However, traditional Y2H system libraries have certain limitations. First, the proteins studied in this system must be located in the nucleus [31,32], and as the intact membrane proteins are connected to the phospholipid bilayer, they will be misfolded when they enter the nucleus. Second, this system is not applicable to proteins that rely on post-translational processing (such as phosphorylation, glycosylation, hydroxylation, disulfide bond formation, and subunit polymerization) for biological activity. It is also not suitable for the screening of membrane-localized proteins. Since membrane proteins exist in the hydrophobic environment of the phospholipid bilayer, if they leave the hydrophobic environment of the membrane, there is a high probability that the tertiary structure of the membrane protein will be changed and its function will be lost and, therefore, the interaction between proteins will not occur [33].

In the current study, we used a split-ubiquitin membrane-based Y2H screening system [34,35,36] to identify proteins physically interacting with the slToc159-1 and slToc159-2 receptors initially. This system is a modified Y2H system specifically designed to identify proteins that interact with membrane proteins and utilize ubiquitin proteins. Ubiquitin is a small and highly conserved protein that marks the degradation of other proteins. It consists of 76 amino acids and has two relatively independent domains located at the N-terminus (Nub) and C-terminus (Cub). In ubiquitin molecules, the artificial transcription factor LEXa-VP16 is first linked to the C-terminal (Cub), and then, the two proteins to be detected are respectively fused with the N-terminal (Nub) and C-terminal (Cub). Once an interaction between the proteins is detected, the ubiquitin is effectively recombined, and the transcription factor Lexa-VP16 is recognized and cleaved by ubiquitin-specific proteases (UBPs). It then becomes a monomer and enters the nucleus, which activates the reporter gene. To extend our analysis of the A domain, we used the G and M domains (excluding the A domain) of slToc159-1 and slToc159-2 as bait to perform one-to-one validation with putative interacting proteins that have been identified from cDNA libraries and for which β-galactosidase (β-gal) activity has been detected. To further investigate whether the A domain has potential precursor protein binding ability, bimolecular fluorescence complementation (BiFC) assays were used to analyze the binding of the precursor protein to three domains (A, G, and M) of the tomato slToc159 protein. Our results enrich the specific precursor protein substrate of tomato slToc159 proteins. In addition, the specificity of the slToc159 protein to the precursor protein was not as strict as hypothesized, and both slToc159-1 and slToc159-2 could recognize photosynthetic and non-photosynthetic proteins. Our data confirm the importance of the previously proposed A domain of slToc159 for the selectivity of different substrates; however, this selectivity is highly variable among different precursor proteins. The BiFC assays showed that the highly non-conserved A domain can interact with precursor proteins and that the fluorescence signal occurs on chloroplasts. Furthermore, transient expression of the A domain fusion with GFP in Arabidopsis protoplasts showed that the A domain contained no membrane targeting signal. Overall, our study expands the knowledge of the recognition of precursor proteins by Toc receptors in tomato chloroplasts and provides a foundation for further research on the binding and transport mechanisms of precursor proteins.

## 2. Results

### 2.1. Generation of NMY51 Strains of S. cerevisiae Expressing Functional slToc159 Receptor Family Bait Proteins

In the split-ubiquitin Y2H library screening, the bait was fused to one end of ubiquitin (Cub) and a transcription activator (LEXa-VP16). The cDNA library was then fused to the other end of the ubiquitin (Nub). The bait–prey interaction resulted in the recombination of ubiquitin and the recruitment of proteases (UBPs) that cleave transcriptional activators and then activate reporter genes in the nucleus (Figure 1A). The PCR-amplified 159-1AGM, 159-1GM, 159-2AGM, and 159-2GM fragments were subcloned into pBT3-SET using the *Sfi*I sites provided in the MCS of the DUAL membrane bait vector. All the bait proteins were preceded by a weak CYC promoter and a STE2 leader sequence corresponding to the N-terminal 13 amino acids of the *S. cerevisiae* STE2 protein for targeting the resulting bait proteins to the yeast endoplasmic reticulum membrane (Figure 1B). Western blot results confirmed the expression of the bait proteins (Figure 1C). To ensure that the bait proteins had been correctly expressed in the split-ubiquitin Y2H system, four bait proteins (slToc159-1AGM, slToc159-1GM, slToc159-2AGM, and slToc159-2GM) were co-expressed with the positive prey control protein pOstI-NubI in yeast strain NMY51 via LiOAc/PEG transformation [18]. NMY51 was initially streaked on YPD media, and SD-drop-out media was used where required. The co-expression of each of the four different bait proteins and the pOstI-NubI reporter prey protein resulted in the activation of reporter genes, which was evidenced by the robust growth of yeast on the highly stringent selective medium (SD-Leu-Trp–His-Ade; Figure 1D), indicating that the bait protein was inserted into the yeast endoplasmic reticulum in the correct way and functioned normally in the split-ubiquitin system. The strength of the bait–prey interaction was confirmed by quantitative β-gal activity assays (Figure 1E). High β-galactosidase activity was observed in the strains with four bait vectors (slToc159-1AGM, slToc159-1GM, slToc159-2AGM, and slToc159-2GM) co-transferring with positive prey vectors (pOstI-NubI). The β-galactosidase activity was very low in the strains that were co-transformed with the bait vector and the empty prey vector or the negative control prey, indicating that none of the bait proteins had significant self-activation characteristics. Each pair of plasmid combinations and their purpose of usage are listed in Table 1. The results show that bait vectors can be directly used to screen cDNA libraries in split-ubiquitin membrane systems.

### 2.2. Screening of Tomato cDNA Library with slToc159-1AGM and slToc159-2AGM Bait Proteins

To identify proteins interacting with the slToc159 family, we constructed a high-quality tomato cDNA library. The library construction method adopted the gateway technology. The library had a complexity of 1.6 × 10^7^ transformants and an insert size of 750–2000 kb, and 100% of vectors contained an insert (Figure 2). Using screening protocol, we identified 41 unique putative proteins interacting with slToc159-1 and slToc159-2 (Table 2 and Table 3); 38 of all 41 prey proteins were identified as plastid proteins by the BLAST function in NCBI and the online prediction of target P [37] and Cell-PLoc [38]. The interaction proteins were grouped according to photosynthetic proteins and non-photosynthetic proteins. Prey vectors isolated from separate and different positive clones may contain identical cDNA inserts. For slToc159-1AGM, we identified 21 unique genes that encode putative interacting preproteins. Of these 21 genes, 14 were associated with photosynthesis and 7 were associated with non-photosynthesis. For slToc159-2AGM, we identified 26 unique genes that encode putative interacting preproteins. Of the 26 genes, 8 were associated with photosynthesis and 18 were associated with non-photosynthesis. Prey proteins related to photosynthesis (Table 2) mainly included (i) photosystem-I-associated protein (i.e., photosystem I subunit O (PSI-O), photosystem I reaction center subunit IV A (PSI-A), photosystem I reaction center subunit III (PSI-III), chlorophyll a-b binding protein CAB11, photosystem I reaction center subunit II (PSI-D)), (ii) photosystem-II-associated protein (i.e., photosystem II 23 kDa protein (PSBP), oxygen-evolving enhancer protein 1 (PSBO), photosystem II subunit S (PSBS), chlorophyll a-b binding protein CP24 (LHCP), chlorophyll a-b binding protein 8 (CAB-8), chlorophyll a/b binding protein Cab-3C, chlorophyll a-b binding protein 7 (CAB7)), (iii) the RuBisCO small subunit (i.e., RBSC-1, RBSC-2A, RBSC-3, RBSC-4), and (iv) the ATP synthase small subunit (i.e., ATP synthase subunit B, ATP synthase delta chain). Non-photosynthesis-related interacting proteins play important roles in various metabolic and regulatory pathways (Table 3), such as (i) biosynthetic pathways (stearoyl-[acyl-carrier-protein] 9-desaturase, 3-oxoacyl-[acyl-carrier-protein] reductase (FabG), thiamine biosynthesis protein (ThiC)), (ii) plastid division, i.e., cell division protein FtsZ and calcium-binding protein, (iii) stress response, i.e., A/B barrel domain-containing protein and RPM1-interacting protein 4, (iv) redox regulation, i.e., peroxiredoxin Q, (v) chloroplast gene expression-related proteins, i.e., multiple organellar RNA editing factor 2, and (vi) photorespiration, i.e., glutamine synthetase (GS2). Notably, among the identified prey proteins, three (calcium-binding protein, serine hydroxymethyl transferase, NADH dehydrogenase [ubiquinone] flavoprotein 14) were described by BLAST or predicted by online localization as mitochondrial-targeting proteins. In addition, three cytoplasmic localized prey proteins (60S ribosomal protein, hop-interacting protein (THI026), heat shock cognate 70 kDa protein (HSP70)) were identified, of which HSP70 functions as a molecular chaperone in the initial recognition of the preprotein with Toc159.

We wanted to further confirm that the interactor identified by slToc159 reflects the recognition of the preprotein, and we examined the interaction of slToc159 with two variants of RBCS3 (with or without the N-terminal transit peptide) (Figure 3), but since the ChloroP for the predicted N-terminal transit peptide had been out of service, we performed an alignment using three small RuBisCO subunits of Arabidopsis (RBCS1B, 2B, 3B) with RBCS3 in tomato and determined the length of the transit peptide of tomato RBCS3 to be 54 aa at the N-terminal. We found that RBCS3, lacking the N-terminal transit peptide, had difficulty interacting with slToc159 (Figure 3), so we determined that the identified interactors represented the interaction of Toc159 with the precursor protein. In particular, three non-photosynthetic proteins localized in the cytoplasm were not included.

### 2.3. Interactions of Toc34-1 and Toc34-2 Prey Proteins with Toc159-1AGM and Toc159-2AGM Bait Proteins

Two different “trimeric” Arabidopsis TOC complexes have been isolated, atToc159/atToc33/atToc75 and atToc132 or 120/atToc34/atToc75, which mainly transport photosynthetic and non-photosynthetic proteins, respectively [39]. The stoichiometric ratio of Toc159/Toc34/Toc75 of the purified pea TOC core complex was 1:4–5:4 [40,41], indicating that there were multiple complexes composed of Toc159 and Toc34 with different homologs or different stoichiometry. However, no other Toc GTPase components were identified in our results, so we wondered whether Toc159 and Toc34 homologs in tomato also have potential interaction ability. Two Toc34 homologs in tomato, slToc34-1 and sltoc34-2, were subcloned into the prey vector pPR3-N and transferred into strains containing Toc159-1 and Toc159-2 via LiOAc/PEG transformation, respectively. As shown in Figure 4, four yeasts co-expressing bait and prey (slToc159-1/slToc34-1, slToc159-1/slToc34-2, slToc159-2/slToc34-1, slToc159-2/slToc34-2) could grow on the highly stringent selective medium (SD-Leu-Trp–His-Ade), indicating that two Toc159 homologs (slToc159-1 and slToc159-2) could interact with two Toc34 (slToc34-1 and slToc34-2) homologs in tomato, respectively. Further β-galactosidase activity assays also verified this result. 

### 2.4. Interactions of Photosynthesis-Related Prey Proteins with Four Bait Proteins (slToc159-1AGM, slToc159-1GM, slToc159-2AGM, slToc159-2GM)

Using slToc159-1AGM and slToc159-2AGM as bait, a total of 18 photosynthesis-related proteins were identified, and 4 of the 18 unique proteins were confirmed by screening with both bait proteins (Table 3). Overall, of the identified photosynthesis-related proteins, most (14/18) were identified by slToc159-1 as bait, suggesting that slToc159-1 or slToc159-2 has different affinities for different types of preproteins. We compared the interaction strength of the photosynthesis-related proteins with each of slToc159-1AGM and slToc159-2AGM and evaluated the possibility that the N-terminal hypervariable A domain of slToc159-1 or slToc159-2 affects the specific recognition of photosynthetic and non-photosynthetic preproteins. Individual pairwise split-ubiquitin Y2H assays were conducted using four bait vectors (slToc159-1AGM, slToc159-1GM, slToc159-2AGM, and slToc159-2GM) co-transformed with prey vectors containing cDNA fragments encoding photosynthesis-related proteins. Qualitative analysis PXG assays and the quantitative X-gal method were used to detect the interaction strength of the bait protein and prey protein in each co-expression clone. The photosynthesis-related proteins identified only with slToc159-1AGM as bait could interact with slToc159-2AGM. The photosynthesis-related proteins identified only with slToc159-2AGM as bait could also interact with slToc159-1AGM. Most photosynthetic prey proteins appear to interact more strongly with slToc159-1 than with slToc159-2, as observed by the PXG method and subsequently demonstrated by the quantification of β-gal activity (Figure 5). Three photosynthetic prey proteins (RBCS4, PSI-O, ASS-B) exhibited similar strengths of interaction with slToc159-1AGM and slToc159-2AGM bait proteins. LHCP showed a stronger interaction with slToc159-2AGM compared to slToc159-1AGM. Truncation of the A domain of slToc159-1AGM in the bait protein (slToc159-1GM) resulted in significantly lower interaction strength with the photosynthetic prey protein compared to slToc159-1AGM containing the A domain based on quantitative β-galactosidase activity assays (Figure 5B), for instance, RBCS-2A (62.4%), RBCS3 (24.5%), PSI-D (62%), PSI-O (93.8%), PSI-III (46.3%), PSBS (76.6%), PSBO (60%), PSBP (88.7%), CAB-3C (38.4%), CAB-7 (65.4%), CAB-8 (49.2%), and CAB-11 (20.7%) (numbers in parentheses indicate decreases in interaction strength). The five interactors (RBCS1, RBCS-4, PSI-A, LHCP, ASS-delta) observed in the assay did not significantly alter the strength of the interaction with slToc159-1, regardless of the presence or absence of the A domain. The β-gal activity assay was also used to compare the same set of photosynthetic prey proteins to two slToc159-2 bait variants, slToc159-2AGM and slToc159-2GM. In contrast to the qualitative PXG analysis of slToc159-1 as a bait protein, the affinity of the six photosynthesis-related interacting proteins (RBCS-1, RBCS2A, PSI-D, PSBO, CAB-3C, CAB-7) to slToc159-2GM was higher than that to slToc159-2AGM (Figure 4C). The quantitative analysis of β-gal showed that the interaction strength increased by 79.9%, 615.4%, 57.8%, 132.8%, 1357.1%, and 21.7%. The affinity of the six photosynthesis-related interacting proteins (RBCS-3, PSI-A, PSI-O, PSI-III, LHCP, and PSBP) to Toc159-2GM decreased by 37.1%, 53.7%, 72.7%, 86%, 75.5%, and 89.2%, respectively, compared to Toc159-2AGM. Collectively, the data here suggest that slToc159-1 has a stronger affinity for photosynthesis-related proteins and that this strong affinity is largely conferred by the A domain. Regarding the affinity of slToc159-2 to the preprotein, the presence of the A domain positively affects some photosynthesis-related proteins but negatively affects the other photosynthesis-related proteins.

### 2.5. Interactions of Non-Photosynthesis-Related Prey Proteins with Four Bait Proteins (slToc159-1AGM, slToc159-1GM, slToc159-2AGM, slToc159-2GM)

Of the 22 unique non-photosynthetic prey proteins identified as interactors from the cDNA libraries, most (15/22) were identified using slToc159-2 as bait. Of all 15 prey proteins uniquely identified in the slToc159-2 AGM bait screening, seven prey proteins (ACP2, ACP4, CDP2-1, HflX, THI026, EF1 and FabG) also interacted with three variants of bait (i.e., slToc159-1GM, slToc159-2AGM, and slToc159-2GM). Seven of twenty-two prey proteins (RBP1, MORF2, RSH1, KIC, RIP4, SLP1, and SHT) could only interact with the two variants of slToc159-2 (slToc159-2AGM and slToc159-2GM). Two prey proteins (SLP1 and SHT) could only interact with slToc159-2AGM. The same test strategy was used to analyze non-photosynthesis-related prey proteins. Comparison of the interaction intensity between non-photosynthetic prey proteins and slToc159-1AGM and slToc159-2AGM (Figure 6B) showed that most non-photosynthetic prey proteins had high affinity for slToc159-2. Among the non-photosynthetic prey proteins, GS2 (28.8%), CDP2-1 (66.1%), SAC9 (866.1%), and ThiC (475.5%) interacted significantly with slToc159-1AGM compared to slToc159-1GM. The numbers in brackets represent the relative increase in the interaction strength of slToc159-1AGM compared to slToc159-1GM. By contrast, the interaction strength of NADH14 with slToc159-1GM was significantly higher than that with slToc159-1AGM. There was no significant difference in interaction intensity between the remaining non-photosynthetic prey with slToc159-1AGM and slToc159-1GM (Figure 6). The interaction strength of the seven prey proteins with slToc159-2AGM was significantly stronger than that with slToc159-2GM, i.e., THI026 (151.7%), HSP70 (405.9%), SAC9 (81.1%), ThiC (81.6%), SLP1 (509.1%), SHT (888.9%), and FabG (61.7%). The remaining non-photosynthetic prey did not differ significantly in interaction strength with slToc159-2AGM and slToc159-2GM. In summary, the data suggest that slToc159-2 has a higher affinity with most non-photosynthesis-related proteins; the presence of the A domain in both slToc159-1 and slToc159-2 alters the affinity for certain non-photosynthesis-related prey proteins.

### 2.6. BiFC Analysis of Preprotein Interactions between Different Domains (-A, -G, -M) of slToc159-1 and slToc159-2

To further explore the domain of interaction between slToc159-1 and slToc159-2 receptors and precursor proteins, we fused different slToc159 variants into the N-terminus of GFP; RuBisCO small subunit 3 (RBCS3) (containing N-terminal 54 amino acid cTP), a representative nuclear-coding chloroplast protein in photosynthesis, was fused to the C-terminal of GFP, and BiFC analysis was performed (Figure 7). First, the results of the two-hybrid splitting of ubiquitin yeast were verified by BiFC experiments, which indicated that slToc159 directly interacts with RBCS3 and is located in the chloroplasts (Figure 7A). slToc159-1 and slToc159-2 were identified as two GTPases, as multiple well-defined GTPase motifs were found in the highly conserved G domain. Given the importance of the G domain, we tested the ability of the G domain to interact with the precursor protein by BiFC and, as expected, the G domain alone was also able to interact with RBCS3 and a diffused fluorescence signal was observed, indicating cytoplasmic localization in tobacco cells (Figure 7B). Subsequently, we tested the binding ability of the A domain and M domain to the proprotein, and we found that only the M domain could not interact with RBCS (Figure 7C), which seemed unnecessary for the recognition function of slToc159 to the preprotein. After the fusion of the G domain and M domain, diffused fluorescence signals could be observed again, indicating the important role of the G domain in precursor protein recognition (Figure 7D). Interestingly, although previous reports on the highly variable A domain of the amino acid sequence suggested that it was independent of the function of Toc159, our results found that the A domain could interact with RBCS3 alone and the fluorescence signal overlapped with the chloroplast’s own fluorescence. This indicates that the interaction between slToc159-1A and RBCS3 is located in the chloroplast (Figure 7E), and a comparison of Figure 6A and 6E shows that domain A contains certain chloroplast-targeting signals.

The same strategy was used to analyze the interaction between different slToc159-2 variants and RBCS3, and similar results were obtained. In contrast to the results obtained for slToc159-1GM, the interaction fluorescence signals of slToc159-2GM and RBCS3 aggregated into a cluster, which may represent insoluble aggregation caused by protein misfolding. However, the general trend was obvious, and it overlapped with the spontaneous fluorescence of the chloroplast.

In order to further determine whether the A domain contains chloroplast membrane targeting signals and to more clearly distinguish whether it is located in the chloroplast or on the chloroplast membrane, we tested the location of the Toc159 full-length sequence and the A domain in the Arabidopsis protoplast. The result was unexpected. The fluorescence signal of the A domain was observed in the cytoplasm (Figure 8), which indicated that the A domain did not contain any membrane-targeted signals. Unfortunately, when we observed the localization of slToc159-1 and slToc159-2, we repeated it many times without observing the fluorescence signal. A possible reason is that the long Toc159 amino acid sequence is difficult to express in the protoplast or that the tertiary structure of the Toc159 protein encapsulates the GFP.

## 3. Discussion

Tomato contains a large number of photosynthetic and non-photosynthetic plastids and also exemplifies the transformation of photosynthetic (chloroplasts) plastids into non-photosynthetic (especially chromoplasts) plastids, which prompted us to study the Toc complex in tomato. We previously identified Toc GTPase homologs in tomato based on the whole tomato genome, which provided a foundation for further study on the functional differences between tomato Toc GTPase homologs. The expression profile of Toc GTPases indicates the functional differences of different Toc in different plastids, and previous studies on Toc159 receptor functions have put forward the hypothesis that different import pathways exist for different functional classes of proteins [39,40,41]. This raises some questions about the function of Toc complexes in tomato, such as whether different Toc GTPase homologs specifically transport specific types of precursor proteins, how multiple Toc components assist each other to lead to the biogenesis of different types of plastids, and whether there is a more complex chloroplast protein transport mechanism in tomato. Identifying the protein interactome for the slToc159 family of chloroplast protein import receptors is a crucial step towards answering these questions. In the current study, we sought to expand our understanding of preprotein recognition and substrate specificity in different TOC members by screening and studying the preprotein substrates of two members of the tomato Toc159 family. The classic Y2H system requires that the detected proteins must exist in the matrix in soluble form, and the interaction between proteins occurs in the yeast nucleus. Considering the particularity of Toc as a membrane protein, it is highly likely to lead to tertiary structure changes and function loss of membrane proteins when it leaves the hydrophobic environment of the membrane. We used a special strategy, namely, a split-ubiquitin Y2H system, to provide a physiological hydrophobic environment for membrane proteins that is more conducive to the detection of interactions between membrane proteins and soluble proteins. Since chloroplast development and biogenesis are active in the seedling stage, we generated cDNA libraries using RNAs isolated from early tomato development to ensure the maximum abundance of genes encoding imported chloroplast proteins in the library.

The split-ubiquitinated Y2H screening with slToc159-1AGM and slToc159-2AGM as bait was completed, and several candidate proteins for interaction were identified. They could be subdivided into photosynthesis-related and non-photosynthesis-related proteins according to their cellular functions. The overlap of slToc159-1 and slToc159-2 in interactions with preproteins was striking, suggesting some functional redundancy between the two. Compared with the large nuclear genome, the chloroplast genome contained only about 120 to 130 genes encoding about 80 proteins, most of which were involved in photosynthesis, transcription, and translation [42]. The remaining 95% of plastid proteins were encoded by nuclear genes, and so, the fate and function of chloroplasts are mainly regulated by nuclear gene information. All 41 proteins identified in this study to be interacting with slToc159 were identified as plastid proteins encoded by nuclear genes by the online prediction of target P and Cell-PLoc or a description in NCBI. This includes proteins that play an important role in preprotein import, RNA processing, protein maturation and degradation, plastid gene expression, RuBisCO assembly, photosystem assembly, thylakoid biogenesis, photorespiration, chloroplast division, fatty acid biosynthesis, and stress response. The intact thylakoid membranes contain proteins and pigment–protein complexes, including chloroplast lipids, photosystem II, the cytochrome b6f complex, photosystem I, and adenosine triphosphate (ATP) synthase [43]. Except for the cytochrome-b6f-complex-related components that were not identified in the library, other proteins related to thylakoid membrane biogenesis were identified. 

We performed a split-ubiquitin Y2H screen to identify 41 interacting partners of the two receptors of Toc159 in tomato. However, approximately 95% (approximately 2000–2500 chloroplast proteins) were encoded by the nucleus and transported by Toc159. Fewer slToc159 interactions were identified in this study, which may be explained by several factors. In plant cells, for example, cytoplasmic partner proteins (AKR2 and 14-3-3) have been proven to work with chloroplast precursor proteins to play an indispensable auxiliary role during the transfer process [44,45,46], but no components or related events in yeast cells exist. These cells, which inevitably lead to a significant portion of the precursor protein interactions in the yeast cell, could not be detected. In addition, the use of heterologous systems (yeast) can lead to significant differences in the strength of protein–protein interactions. The weak heterodimer events of the basic leucine zipper (bZIP) protein in the plant system cannot be detected in the yeast system [47], suggesting that there are important protein interaction factors in some plant systems. This prevents weaker interactions from being detected in the yeast system. It should not be ignored that the addition of 3-AT to the screening medium to minimize the leakage expression of the His gene probably inhibited the relatively weak interaction between Toc159 and some precursor proteins. Most importantly, the recognition and interaction of the precursor protein with Toc159 are transient and reversible; that is, if the precursor protein continues to recognize Toc receptors in heterogeneous systems, it is inevitable that some of the precursor protein interactions with slToc159 will be missed under our stringent screening conditions. In summary, the low number of identified prey proteins was largely due to the lack of plant-specific factors necessary for chloroplast protein transport in yeast and the omission of the detection of transient interactions between proteins under strict screening conditions. In addition, the technical reasons are non-negligible because the prey library provided a representation of only the most abundant mRNAs. Moreover, that hypothesis-based approach allowed us to confirm the interaction of slToc159 and slToc34 (Figure 2), which was not detected in the screening experiment. It could also be considered that slToc159 expressed in yeast cells could have strong interactors in the cells that outcompete some prey library interactors.

The results of our cDNA library screening showed that both slToc159-1 and slToc159-2 could screen photosynthetic and non-photosynthetic chloroplast proteins as interaction partners. This is not as stringent as our previous predictions that slToc159-1 and slToc159-2 interact exclusively with a specific set of preproteins. Of the identified photosynthesis-related proteins, a large proportion was identified using only slToc159-1 as bait. Among the identified non-photosynthesis-related proteins, a large proportion was identified using only slToc159-2 as bait. Qualitative and quantitative analysis of β-galactosidase activity showed that slToc159-1 had a stronger affinity for photosynthesis-related proteins than slToc159-2. Our previous study also found that the tissue-specific expression levels of the tomato slToc159 GTPase receptor varied widely (for example, the expression level of slToc159-1 was highest in green and photosynthetic tissues (such as the young leaves, green fruits, and flower buds), low in the roots, and medium in the cotyledons and red fruits). This expression pattern is similar to that of atToc159 in Arabidopsis [22], and it is expressed in different tissues. Our results clearly point to the conclusion that slToc159-1 is more focused on transport with photosynthetic precursor proteins and slToc159-2 is more focused on transport with non-photosynthetic precursor proteins. Although our results suggest that different slToc159 receptors mediate the import of different functional proteomes, it will be challenging to determine the exact TOC complex transport pathway for each substrate.

Similar to Arabidopsis thaliana, the sequence similarity of Toc159 family member domains in tomato differed, and the sequences conserved in the A domain were significantly lower than those of the G and M domains [44]. Due to the variable sequence length and the diversity of amino acid sequences in family members, the A domain is hypothesized to confer different substrate recognition specificities. To verify this hypothesis, we constructed two bait proteins that did not contain the A domain of slToc159-1 and slToc159-2 (containing only G and M domains) and conducted one-to-one tests with the identified prey proteins in pairs. The interaction strength of each pair was analyzed by β-gal qualitative and quantitative methods. Of the 18 photosynthesis-related proteins studied, 12 (67%) had significantly stronger affinities to slToc159-1AGM than slToc159-1GM, and the remaining 6 (33%) had an affinity with both bait proteins at higher levels and did not show significant changes. Of the 18 photosynthesis-related proteins studied, 12 (67%) had significantly stronger affinities to slToc159-1AGM than slToc159-1GM, and the remaining 6 (33%) had an affinity with both bait proteins at higher levels and showed no significant changes. Six (33%) showed no significant change in affinity to both bait proteins, and notably, six (33%) had a significantly stronger affinity to slToc159-2GM than slToc159-2AGM, indicating that the presence of the A domain of slToc159-2 appears to have negatively impacted its interactions with certain photosynthesis-related proteins. Among the 22 non-photosynthetic proteins studied, slToc159-1 and slToc159-2 showed a certain degree of specificity, and they each specifically transported a certain non-photosynthetic protein. There was no statistically significant difference in the interaction intensity of some non-photosynthetic proteins, and eight (36%) proteins had significantly higher interaction intensity with slToc159-2AGM than slToc159-2GM. Our results indeed illustrate that the variable A domains of members of the slToc159 family confer specificity to their interaction with different preproteins. We found that slToc159-1 and slToc159-2 recognized both photosynthetic and non-photosynthetic proteins and had a high degree of functional redundancy in their interactions with photosynthesis. This result is consistent with previous proteomic and transcriptome analyses of ppi2 mutants and atToc159 co-inhibitory lines [48], which also indicate the presence of multiple chloroplast protein import pathways. The A domain of slToc159-1 had a positive effect on the recognition of slToc159-1 and photosynthetic proteins, and the A domain of slToc159-2 had a negative effect on the recognition of some photosynthetic proteins, while the A domain of slToc159-1 had no significant effect on the recognition of non-photosynthetic proteins. Similarly, the A domain of atToc159 has been shown to confer specificity on different receptors to recognize different types of precursor proteins. Non-photosynthetic precursor cell proteins bind to receptors with certain specificity, and some can only interact with slToc159-1 or slToc159-2 alone, indicating that different GTPase receptors mediate different non-photosynthetic precursor cell protein imports. It is possible that the specific import pathway of these non-photosynthetic preproteins is not caused by the A domain and that the signal leading to this specific pathway is present in the transit peptides of the precursor proteins themselves.

Earlier studies have shown that the conserved G domain includes at least a portion of transit peptide binding sites [16,26] and that the G domain contains multiple typical GTPase motifs [49] that act as switches regulating precursor protein recognition. However, relatively few studies have focused on the A and M domains, and owing to the lack of known conserved functional motifs in the A domain, the A domain is not considered essential for the slToc159 recognition of precursor proteins. Our previous research found that owing to the diversity of amino acid sequences in the A domain, it may play an important role in the specific recognition of the slToc159 receptor as well as different precursor proteins, and the proportion of the A domain in the overall structure of Toc159 (45% of slToc159-1, 35% of slToc159-2) indicates that it plays an important role in preprotein recognition. Since BiFC is not suitable for detecting a large number of protein interactions, we used only one representative plastid protein, RBCS3, encoded by nuclear genes to analyze interactions with different slToc159 family variants and the photosynthesis-related representative gene RBCS3 to further illustrate the importance of the A domain for preprotein recognition. The BiFC assay was used to study the interaction between different structural variations of slToc159-1 and slToc159-2. BiFC analysis of slToc159-1 and slToc159-2 showed that a single A domain could interact with RBCS3, and the interaction occurred on chloroplasts, indicating that the A domain did contain binding sites for preproteins. The fluorescence signal emitted by the complete slToc159-1 structure indicates that it is located in the chloroplast and that the A domain is not included in slToc159-1; the fluorescence signal appeared in the cytoplasm, which indicates that there may be a chloroplast-targeting signal that exists in the A domain. Although a recent review indicated that there are novel targeting signals and pathways (an N-terminal TP and a reverse TP-like sequence at the C-terminus) in chloroplast proteins [50], several studies on the targeting of the TOC159 GTPase receptor family support the membrane localization ability of the C-terminal M domain. Further observation of the localization of the A domain in the protoplasts of Arabidopsis thaliana shows that the A domain does not contain chloroplast-targeting signals [51,52]. However, more precise binding sites of precursor proteins and specific regions containing chloroplast membrane targeting signals need to be further studied. Taken together, our results demonstrated the important function of the A domain in preprotein recognition.

Many types of plastid biogenesis exist in tomato, including a transformation ability between different plastid types, which involves changes in a large number of pigments, changes in various metabolic pathways, and changes in the plastid internal structure [53,54,55]. All these processes require the expansion and reconstruction of the plastid proteome. Recently, a study confirmed that slSP1 action may promote plastid proteome changes through TOC recombination, resulting in a faster fruit-ripening process [55]. It is well known that TOC receptors exist in different subtypes capable of forming substrate-specific transporters and substrate-specific protein import pathways [56,57]. The decline of photosynthesis-related proteins during tomato fruit development implies the need to reconstitute various isoforms of TOC to accommodate different preproteins (for example, those involved in carotenoid synthesis, lipid metabolism, and chlorophyll catabolism), and a more complex plastid protein transport pathway must exist in tomato than in previously studied Arabidopsis and pea (which do not contain chromosomes). Overall, our current data corroborate previous findings that the A domain of Arabidopsis Toc132 mediates substrate-specific recognition [18], thus supporting that the A domains of slToc159-1 and slToc159-2 in tomato influence their different functions and affinities with precursor proteins. Our screening of putative transport substrates for Toc159 also provides a first insight into the chloroplast-targeted proteome in tomato. In the future, it will be interesting to construct cDNA libraries of different tomato fruit developmental stages to fully understand the dynamic changes in the imported proteome during Toc159-mediated plastid transformation. The results of this study will help elucidate the different functions of the Toc apparatus in tomato as well as lay the foundation for revealing the complex plastid protein transport network in tomato.

## 4. Materials and Methods

All methods were performed according to the relevant guidelines and regulationsStrains and growth media

The Saccharomyces cerevisiae strain NMY51 was used during the screening process (MATahis3D200 trp1-901 leu2-3,112 ade2 LYS2::(lex-Aop)4-HIS3 ura3::(lex-Aop)8-lacZ ade2::(lex-Aop)8-ADE2 GAL4). Yeast cells were grown using standard microbial techniques and media. The cells were cultured at 30 °C to the exponential phase (OD600 nm = 1.5–2) in rich YPDA medium. Media designations are as follows: YPDA is yeast extract-peptone-dextrose plus adenine medium; SD is synthetic-defined dropout (SD-drop-out) medium. Minimal dropout media are designated by the constituent that is omitted (e.g., –leu, –trp, –his, and –ade media lacks leucine, tryptophan, histidine, and adenine). Recombinant plasmids were introduced into *S. cerevisiae* yeast (NMY51) via LiOAc/PEG transformation.

3.Tomato cDNA library construction

Tomato leaves were frozen and ground into powder in liquid nitrogen and transferred to several RNase-free 1.5-mL centrifuge tubes, with the same amount collected in each tube. One milliliter Trizol extract (Invitrogen, Carlsbad, California, USA) was added to each centrifuge tube. After shaking and mixing, the leaves were placed at room temperature for 5–10 min. The subsequent operations followed the reagent instructions of the Trizol total RNA extraction solution. The mRNA was isolated and purified according to the instructions of the Oligotex mRNA Midi Kit (Invitrogen, USA). The quality and concentration of the RNA were measured by a Nanodrop 2000 UV spectrophotometer, and the integrity of the RNA was detected by 1% agarose gel electrophoresis. According to the instructions of the Clone Miner II cDNA Library Construction Kit (Invitrogen, USA), the isolated and purified mRNA was back-transcribed into double-stranded cDNA. The double-stranded cDNA was flatly terminated with T4 DNA polymerase. The cDNA was connected to a three-frame attB1 recombinant connector (one for each of the three types of connectors). All the ligation products were loaded into cDNA size fractionation columns, the cDNA fragments were separated according to the instructions, and double-stranded cDNA between 750 and 2000 bp was collected. The collected double-stranded cDNA was ligated to the pDNOR222 vector by constructing a BP recombinant reaction. The recombinant product was transformed into *Escherichia*
*coli* DH10B competent cells (Beijing Bomide Gene Co., Ltd., Beijing, China). After the transformation, 4 mL SOC medium was added and cultured at 37 °C with 225–250× *g* shaking for 1 h. After culturing, the primary library liquid was obtained and the library capacity was identified. The remaining cultures were combined with 40% glycerol at a ratio of 1:1, which constituted the primary library bacterial fluid. The plasmid was extracted from the identified primary library and constructed LR reaction. The uncut library plasmid was recombined with pPR3-N-DEST (16 °C, overnight) and transferred into *E.*
*coli* DH10B receptor cells to obtain the secondary library liquid for evaluating library quality. The remaining cultures were combined with 40% glycerol at a ratio of 1:1, which constituted the secondary library bacterial fluid. The secondary library plasmids were extracted from the bacterial liquid and used for Y2H cotransformation. A total of 2000 clones were obtained on a LB medium plate (containing ampicillin). The titer of the library was 4.0 × 10^6^ CFU·mL^−1^, and the library capacity was 1.6 × 10^7^ CFU. Twenty-four monoclones were randomly selected for colony PCR detection. 

4.Construction of bait vectors expressed in yeast

All bait vectors were constructed using a Cub-LexA-VP16 domain-based vector (pBT3-SET) supplied by the manufacturer (Dual systems Biotech, Zurich, Switzerland). The corresponding amino acid sequences of 159-1 and 159-2 were amplified using the full-length sequences previously cloned in our laboratory as templates. The domains of the tomato slToc159 chloroplast preprotein receptors (slToc159-1AGM, slToc159-1GM, slToc159-2AGM, and slToc159-2GM) were cloned into the provided bait vector pBT3-STE, respectively. For the slToc159-1GM bait construct, a PCR clone corresponding to amino acids 636–1403 of slToc159-1AGM was cloned. Similarly, the bait proteins for slToc159-2GM were generated through the cloning of cDNA fragments of slToc159-2AGM-encoding amino acids 410–1162. All inserts used the SfiI restriction endonuclease site, which was designed to fuse Cub (C-terminus of the yeast ubiquitin, amino acids 34–76) and LexA-VP16 transcription factor coding sequences into the C-terminal of the cDNA fragments. The PCR fragment should start with the first codon after the ATG codon of the open reading frame (ORF), and the stop codon in the PCR fragment must be removed to ensure a continuous translation from the upstream SUC2 ORF and the downstream Cub-LexA-VP16 ORF. It was confirmed by sequencing that the inserted fragment was in the frame with the N-terminal STE2 and C-terminal CUB-LEXa-VP16 without mutation.

5.Split-ubiquitin Y2H screening

The *S. cerevisiae* strain NMY51 was transformed with slToc159-1AGM or slToc159-2AGM bait constructs, and a split-ubiquitin Y2H assay was performed, utilizing positive, negative, and empty prey plasmids; 7 mg of library DNA plasmid was used for each screen, with transformation efficiencies above 2 × 10^6^ clones/mg DNA for all of the library transformations. Owing to a leaky HIS3 gene expression, the bait plasmids and library plasmids pPR3-N were pre-screened to determine the concentration of the *His* gene product inhibitor 3-aminotriazole (3-AT) at 20 mM for the slToc159-1AGM bait and 30 mM for the slToc159-2AGM bait. The library plasmid was transformed into yeast strain NMY51, containing the slToc159-1AGM or Toc159-2AGM bait plasmid. The transformed product was coated on a selective medium (SD-LWHA) plate containing 3-AT, according to the DUAL hunter system instructions. However, perhaps due to the strict screening conditions, there was no clone growth on the selective medium (SD-LWHA), so we reduced the screening intensity. The transformed products were first coated on the medium (SD-LW) plate and cultured at 30 °C for 3–5 days until white clones had grown, and then, the clones were streaked on the selective medium (SD-LWHA) plate containing 3-AT. A clone that is repeated three times and still grows can be considered an interactor. From those interactors grown on SD-LWHA/3-AT selective medium for 3–4 days, prey particles were isolated, transformed into *E. coli*, and re-isolated. The isolated prey and bait protein interactions were further validated by pairwise Y2H interaction studies using slToc159-1 and slToc159-2 as bait and each individual prey. The prey plasmids carrying an insert were sequenced and identified through BLAST.

6.Immunoblot analysis

Western blot was performed by the wet transfer method. Total protein was extracted from yeast cells. After 8% polyacrylamide gel electrophoresis, the protein was transferred to the PVDF membrane, and then, the PVDF membrane was blocked with 5% skim milk at room temperature for 2 h. The PVDF membrane was incubated with a murine monoclonal anti-Lexa antibody as a primary antibody at room temperature for 2 h. After washing with 1 × PBST buffer solution (containing 0.05% Tween-20) four times, the goat anti-mouse was incubated for 1 h. Finally, the membrane was incubated with peroxidase HRP coloration solution and exposed to a Tanon 5200 chemiluminescence detector for protein signal detection (Shanghai Tieneng Technology Co., Ltd., Shanghai, China).

7.β-Galactosidase activity assay

Methods for the qualitative and quantitative determination of the β-gal activity of *S.*
*cerevisiae* extracts have been described previously [58] and were based on instructions from the yeast protocol handbook (Clontech, Mountain View, CA, USA).

8.O-nitrophenyl-ß-d-galactopyranoside (ONPG) assay

One absorbance unit of yeast cells per sample was used. Cells were pelleted in Eppendorf^®^ tubes and resuspended in 500 μL Z-buffer. Ten microliters of chloroform and fifteen microliters of 0.2% sodium dodecyl sulfate (SDS) were added, and the tubes were vortex-mixed briefly. After a 5 min incubation at 28 °C, 100 μL of ONPG solution was added. The reaction was stopped after 25 min for all samples by adding 250 μL of chilled 1 M Na_2_CO_3_. Absorbance was determined at 420 nm.

9.Pellet X-gal (PXG) assay

For each interaction pair, several colonies were picked from the selection plates and inoculated into snap-cap tubes containing 5 mL of selective medium. Cultures were grown from an absorbance (A)546 of <0.1 to an A546 of 0.8–1. One absorbance unit of yeast was pelleted by centrifugation at 2000× *g* for 5 min. The supernatant was discarded, and cell lysis was performed by two freeze–thaw cycles (3 min in liquid nitrogen, 3 min in a 37 °C water bath). Pellets were subsequently resuspended in 20 μL of water, transferred to a transparent flat-bottom 96-well microplate, mixed with 100 μL of phosphate-buffered saline (PBS) buffer (pH 7.4, containing 500 μg/mL X-gal (Applichem, Darmstadt, Germany)), 0.5% (*w*/*v*) agarose, and 0.05% (*v*/*v*) β-mercaptoethanol, and incubated at room temperature.

10.BiFC analysis

The full-length coding sequences of slToc159-1 and RBCS3, without stop codons, were inserted into the pc1300-GN and pc1300-GC (GN and GC) vectors, respectively. In addition, to map the interaction domains, the N-terminal of slToc159-1 (aa 1-635, slToc159-1A), the middle fragment of slToc159-1 (aa 635-998, slToc159-1G), and the C-terminal of slToc159-1 (aa 998-1403, slToc159-1M) were inserted into the GN vector to verify their possible interaction with RBCS3. Similarly, the full-length coding sequences of slToc159-2 and RBCS3, without stop codons, were inserted into the pc1300-GN and pc1300-GC (GN and GC) vectors, respectively. In addition, to map the interaction domains, the N-terminal of slToc159-2 (aa 1-409, slToc159-2A), the middle fragment of slToc159-2 (aa 409-745, slToc159-2G), and the C-terminal of slToc159-2 (aa 745-1162, slToc159-2M) were inserted into the GN vector to verify their possible interaction with RBCS3. The resulting constructs were transferred into GV3101 and subsequently used to transform 5-week-old *Nicotiana*
*benthamiana* leaves. After 48–96 h, green fluorescent protein (GFP) was visualized via a confocal laser scanning microscope (ZEISS, LSM780, Oberkochen, Germany).

11.Subcellular localization analysis

To produce the GFP fusion constructs for subcellular localization analysis, the A domain coding sequences of slToc159-1 and slToc159-2, without stop codons, were inserted into plant expression vector pEGOEP35S to provide a C-terminal GFP. The protoplasts were prepared from the leaves of well-grown wild-type Arabidopsis thaliana that had been growing for about two weeks. The GFP empty vector and fusion expression vector were respectively introduced into the protoplasts by the transient transfection method and incubated for 16 h at room temperature under darkness. The localization of the A domain fusion protein was determined via a confocal laser scanning microscope (ZEISS, LSM780, Germany).

12.Statistical analysis

Statistical analysis of the data was calculated using Student’s *t*-test at *p* < 0.05.

## Figures and Tables

**Figure 1 plants-11-02923-f001:**
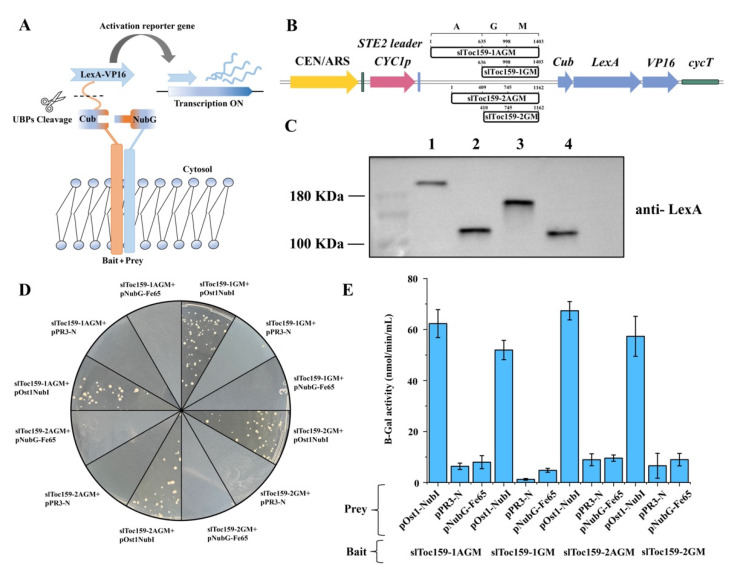
Analysis of slToc159-1AGM, slToc159-1GM, slToc159-2AGM, and slToc159-2GM bait plasmids transformed into NMY51. (**A**) Split-ubiquitin yeast two-hybrid mechanism. (**B**) Diagrammatic representation of the domain organization of the slToc159-1AGM, slToc159-1GM, slToc159-2AGM, and slToc159-2GM bait constructs in the yeast plasmid, pBT3-STE. The bait vector provides an upstream yeast STE2 leader sequence and yeast ubiquitin Cub (34–76 aa), LexA, and VP16 genes downstream. Fusion proteins produced by this cassette are expressed constitutively by the yeast CYC1 promoter and terminator. The bait protein domains were cloned directionally (using SfiI) into the position indicated. The numbers refer to the amino acid sequence of atToc159 or atToc132. (**C**) Immunoblot analysis of whole cell extracts of NMY51 yeast strains expressing the bait proteins. 1: slToc159-1AGM; 2: slToc159-1GM; 3: slToc159-2AGM; 4: slToc159-2GM. (**D**) The split-ubiquitin membrane-based yeast two-hybrid analysis confirming the expression of the bait proteins. The slToc159-1AGM, slToc159-1GM, slToc159-2AGM, or slToc159-2GM bait protein was co-expressed in the *S. cerevisiae* strain NMY51 with the positive prey construct pOst1-NubI, empty prey vector pPR3-N, or the non-interacting negative control construct pNubG-Fe65 and assayed on quadruple selective media (SD-LWHA) plates. Strains co-expressing bait protein and positive control prey exhibited growth only on SD-LWHA selective media. (**E**) Quantitative b-galactosidase activity assay.

**Figure 2 plants-11-02923-f002:**
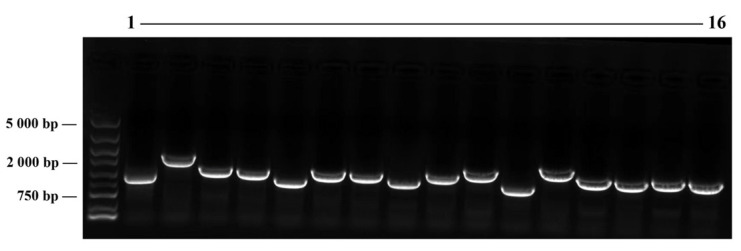
Identification of insert length and recombination rate of cDNA libraries. 1–16: sixteen monoclones were randomly selected for colony PCR detection, and the PCR products were detected by 1% agarose gel electrophoresis.

**Figure 3 plants-11-02923-f003:**
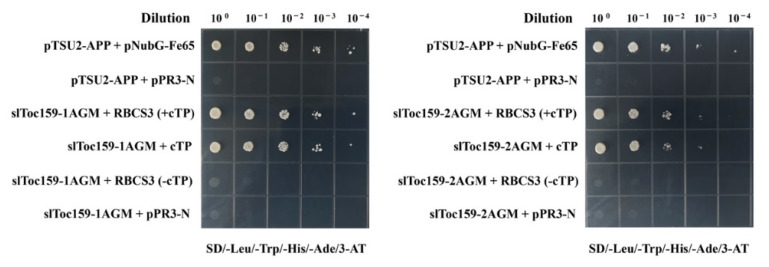
Pairwise yeast two-hybrid interaction studies using slToc159 as bait, with RBCS3 (with or without cTP) as prey. Names in the figure represent the bait protein/construct and the prey protein/construct, respectively. RBCS3(+cTP) represents full-length structures containing transport peptides as prey protein; RBCS3(-cTP) represents truncated structures that do not contain postal transport peptides as prey protein. pTSU2-APP and pNubG-Fe65 as positive controls; pTSU2-APP and pPR3-N as negative controls (Dual systems Biotech). Before photography, the bacterial solution was spot-connected on a petri dish and incubated at 30 °C for 3 days (SD-LWHA/3-AT).

**Figure 4 plants-11-02923-f004:**
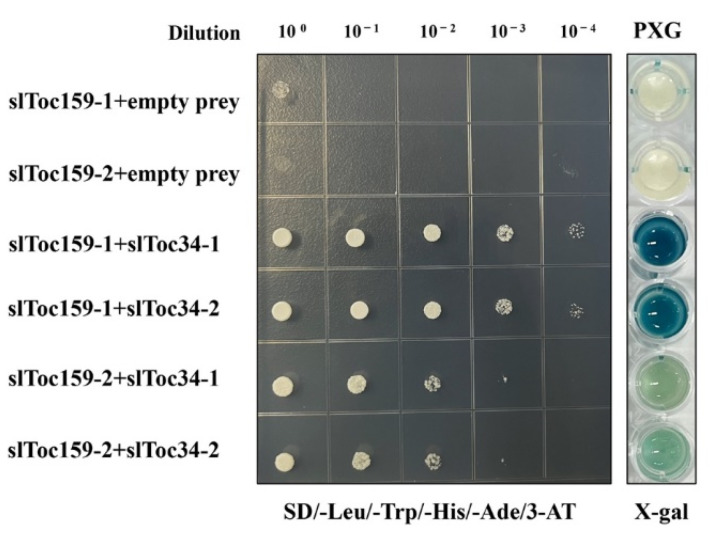
Pairwise yeast two-hybrid interaction studies using slToc159 as bait with Toc34 as prey and the qualitative assay of the reporter gene LacZ using X-Gal. Using slToc34-1 and slToc34-2 as prey proteins and slToc159-1 and slToc159-2 as bait proteins, they were co-transformed in pairs and co-expressed in Saccharomyces cerevis strain NMY51.SD-LW) and selective protein interaction medium (i.e., SD-LWHA/3-AT with 10 mm 3-aminotriazole quadruple selective medium). Before photography, the bacterial solution was spot-connected on a petri dish and incubated at 30 °C for 6 days (SD-LWHA/3-AT). A PXG assay of the reporter gene LacZ using X-Gal as a substrate.

**Figure 5 plants-11-02923-f005:**
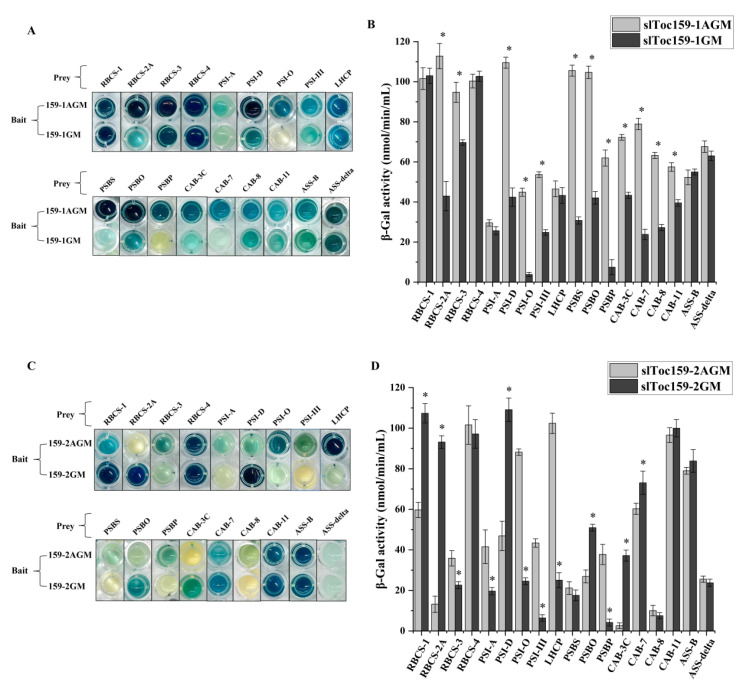
Interactions of photosynthesis-related prey proteins with four bait proteins (slToc159-1AGM, slToc159-1GM, slToc159-2AGM, slToc159-2GM) using the PXG and ONPG assays. (**A**,**C**) Independent transformants were assayed per protein interaction, scanned image of a PXG assay in a 96-well microtiter plate after 60 min of incubation. (**B**,**D**) Quantitative b-galactosidase activity assay. Strains co-expressing respective bait and prey constructs were used in a 96-well microtiter plate-based β-galactosidase assay using (X-Gal) as a substrate. The graph shows β-galactosidase activity/mL per min measured after 60 min. The values represent the mean of three independent experiments. β-gal, β-galactosidase; PXG, pellet X-gal; ONPG, O-nitrophenyl β-d-galactopyranoside; X-gal, 5-bromo-4-chloro-3-indolyl-β-d-galactoside. Values marked with asterisks are significantly different (Student’s *t*-test; *p* ≤ 0.05).

**Figure 6 plants-11-02923-f006:**
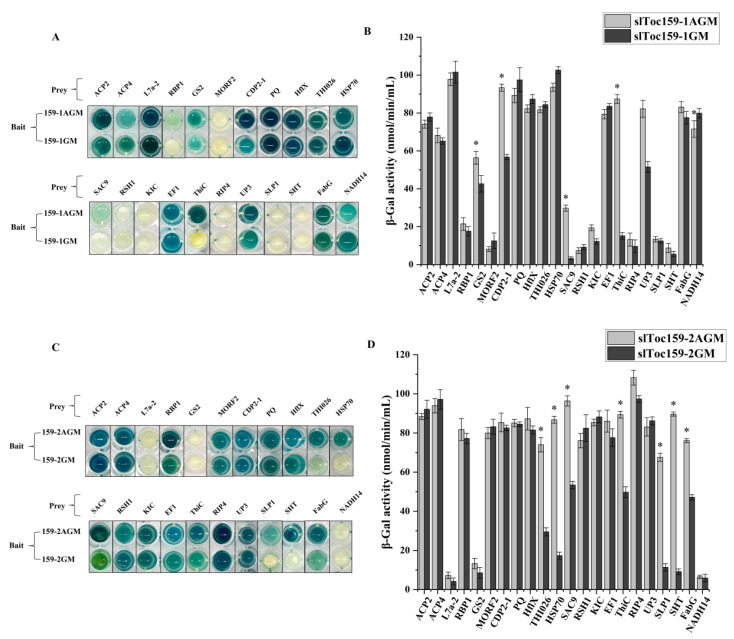
Interactions of non-photosynthesis-related prey proteins with four bait proteins (slToc159-1AGM, slToc159-1GM, slToc159-2AGM, slToc159-2GM). (**A**,**C**) Independent transformants were assayed per protein interaction, scanned image of a PXG assay in a 96-well microtiter plate after 60 min of incubation. (**B**,**D**) Quantitative b-galactosidase activity assay. Strains co-expressing respective bait and prey constructs were used in a 96-well microtiter plate-based β-galactosidase assay using (X-Gal) as a substrate. The graph shows β-galactosidase activity/mL per min measured after 60 min. The values represent the mean of three independent experiments. Values marked with asterisks are significantly different (Student’s *t*-test; *p* ≤ 0.05).

**Figure 7 plants-11-02923-f007:**
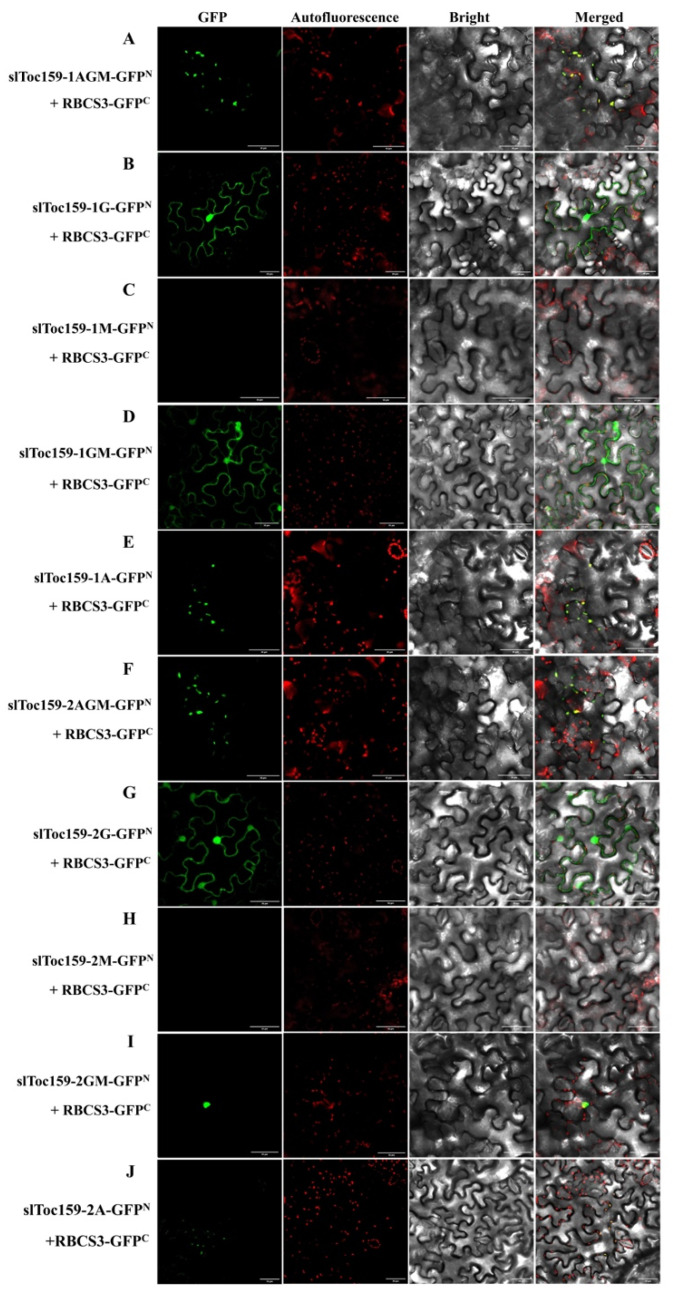
The interaction of different structural variants of slToc159-1 and slToc159-2 with RBCS3 in tobacco leaf epidermal cells. GFP, GFP fluorescence; Autofluorescence, chloroplast autofluorescence; Bright, bright-field image; and Merged, the merge of GFP, Autofluorescence. Bars = 50 μm.

**Figure 8 plants-11-02923-f008:**
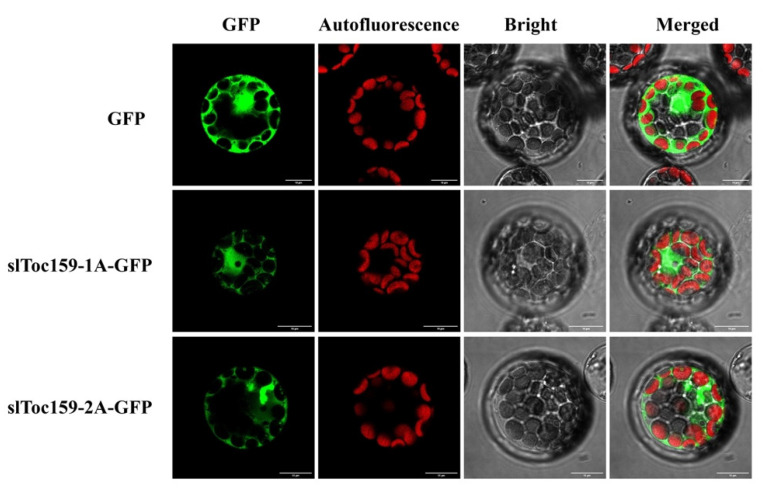
Confocal microscopy images of Arabidopsis thaliana leaf protoplasts transiently expressing slToc159–1A–GFP and slToc159–2A–GFP. The GFP fluorescence signal is shown in green, and chlorophyll autofluorescence is shown in red. Bright-field images confirm the intactness of the protoplasts. An overlay of these three is also shown (Merge). Bars =10 μm.

**Table 1 plants-11-02923-t001:** The combination of plasmid and usage.

Plasmid Combination	Usage
Bait + pOst1-NubI	Validating the correct expression of yeast proteins in the split-ubiquitin system
Bait + pPR3-N Bait + pNubG-Fe65	Self-activation detection

**Table 2 plants-11-02923-t002:** Photosynthesis-related proteins identified to interact with slToc159-1 and slToc159-2 in a split-ubiquitin yeast two-hybrid screen.

		Bait ^2^		Speculative Function	Location
Gene Name	Gene Code ^1^	Toc159-1AGM	Toc159-2AGM		
RuBisCO small subunit 1(RBSC-1)	Solyc02g063150.2	+		Carbon fixation	Chloroplast
RuBisCO small subunit 3 (RBCS-3)	Solyc01g073930.3.1		+	Carbon fixation	Chloroplast
RuBisCO small subunit 2A (RBCS-2A)	Solyc03g034220.3.1	+		Carbon fixation	Chloroplast
Photosystem II 23 kDa protein (PSBP)	Solyc07g044860.3.1		+	Photosystem-II-associated protein	Chloroplast
Photosystem I subunit O (PSI-O)	Solyc06g074200.4.1	+	+	Photosystem-I-associated protein	Chloroplast
RuBisCO small subunit 4 (RBCS-4)	Solyc02g085950.4.1	+	+	Carbon fixation	Chloroplast
Oxygen-evolving enhancer protein 1 (PSBO)	Solyc02g065400.3.1	+		Photosystem-II-associated protein	Chloroplast
Photosystem I reaction center subunit IV A (PSI-A)	Solyc06g083680.3.1		+	Photosystem-I-associated protein	Chloroplast
Photosystem I reaction center subunit III (PSI-III)	Solyc02g069460.2	+		Photosystem-I-associated protein	Chloroplast
photosystem II subunit S (PSBS)	Solyc06g060340.3.1	+		Photosystem-II-associated protein	Chloroplast
Chlorophyll a-b binding protein CP24 (LHCP)	Solyc01g105030.2	+		Photosystem-II-associated protein	Chloroplast
Chlorophyll a-b binding protein 8 (CAB-8)	Solyc10g007690.2	+		Photosystem-II-associated protein	Chloroplast
Chlorophyll a/b binding protein Cab-3C (CAB-3C)	Solyc03g005780.1	+		Photosystem-II-associated protein	Chloroplast
Chlorophyll a-b binding protein (CAB11)	Solyc03g115900.2	+	+	Photosystem-I-associated protein	Chloroplast
Chlorophyll a-b binding protein 7 (CAB-7)	Solyc10g006230.2		+	Photosystem-II-associated protein	Chloroplast
Photosystem I reaction center subunit II(PSI-D)	Solyc06g054260.1.1	+		Photosystem-I-associated protein	Chloroplast
ATP synthase subunit B (ASS-B)	Solyc06g066000.3.1	+	+	ATP synthesis driven by the proton dynamic potential	Chloroplast
ATP synthase subunit delta chain (ASS-delta)	Solyc05g050500.1	+		ATP synthesis driven by the proton dynamic potential	Chloroplast

^1^ Gene code from the GenBank database or Sol Genomics Network (SGN). ^2^ ‘+’ indicates which bait protein the prey protein interacts with.

**Table 3 plants-11-02923-t003:** Non-photosynthesis-related proteins identified to interact with slToc159-1 and slToc159-2 in a split-ubiquitin yeast two-hybrid screen.

		Bait ^2^		Speculative Function	Location
Gene Name	Gene Code ^1^	Toc159-1AGM	Toc159-2AGM		
ATP-dependent Clp protease proteolytic 2 (ACP2)	Solyc08g079620.2		+	A central component of the chloroplast protease network	Chloroplast
ATP-dependent Clp protease proteolytic 4 (ACP4)	Solyc08g077890.2		+	A central component of the chloroplast protease network	Chloroplast
60S ribosomal protein (L7a-2)	Solyc06g064470.4.1	+		Ribosome biogenesis	Cytoplasm
Glycine-rich RNA-binding protein (RBP1)	Solyc01g109660.2		+	Development and stress adaptation	Chloroplast
Glutamine synthetase (GS2)	Solyc01g080280.2	+		Photorespiration and assimilation of ammonia from nitric acid reduction	Chloroplast
Multiple organellar RNA editing factor 2 (MORF2)	Solyc06g008220.2		+	Multiple RNA editing in plastids	Chloroplast
Cell division protein FtsZ (CDP2-1)	Solyc09g009430.3.1		+	Key cellular skeletal components in the mechanism of chloroplast division	Chloroplast
Peroxiredoxin Q (PQ)	Solyc07g042440.2	+	+	Redox reactions	Chloroplast
GTPase (HflX)	Solyc04g080770.3.1		+	GTP enzymes that target chloroplasts	Chloroplast
Hop-interacting protein (THI026)	Solyc04g018110.1.1		+	Assemble the Hsp complex	Cytoplasm
Heat shock cognate 70 kDa protein (HSP70)	Solyc11g066060.3.1	+	+	Assists in targeted transport of preproteins to chloroplasts	Cytoplasm
Stearoyl-[acyl-carrier-protein] 9-desaturase (SAC9)	Solyc03g063110.2	+		Fatty acid metabolism	Chloroplast
GTP diphosphokinase (RSH1)	Solyc09g098580.3.1		+	Transfer of high-energy phosphate groups and signal transduction	Chloroplast
Calcium-binding protein (KIC)	Solyc10g009340.1.1		+	Regulation of cell division and trichome morphogenesis	Mitochondrion
Elongation factor 1-alpha (EF1)	Solyc06g005060.3.1		+	Regulation of cell growth and division	Chloroplast
thiamine biosynthesis protein (ThiC)	Solyc06g006080.3.1		+	Vitamin B1 biosynthesis	Chloroplast
RPM1-interacting protein 4 (RIP4)	Solyc06g083390.4.1		+	Plant immunity	Chloroplast
A/B barrel domain-containing protein (UP3)	Solyc07g041490.1	+	+	Stress response	Chloroplast
SufE-like protein 1 (SLP1)	Solyc12g015910.1.1		+	Formation of iron-sulfur clusters in chloroplasts	Chloroplast
Serine hydroxymethyl transferase (SHT)	Solyc05g053810.2		+	Catalyze the conversion of serine and glycine to each other	Mitochondrion
3-oxoacyl-[acyl-carrier-protein] reductase (FabG)	Solyc12g045030.1		+	Fatty Acid Synthesis	Chloroplast
NADH dehydrogenase [ubiquinone] flavoprotein 14	Solyc02g087240.2	+		Redox reactions	Mitochondrion

^1^ Gene code from the GenBank database or Sol Genomics Network (SGN). ^2^ ‘+’ indicates which bait protein the prey protein interacts with.

## Data Availability

The amino acid sequences analyzed in the current study are available in the Sol Genomics Network repository (https://solgenomics.net/, accessed on 20 July 2022) and the NCBI database (https://www.ncbi.nlm.nih.gov/, accessed on 20 July 2022).

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
