# Peer review of "Split-Ubiquitin Two-Hybrid Screen for Proteins Interacting with slToc159-1 and slToc159-2, Two Chloroplast Preprotein Import Receptors in Tomato (Solanum lycopersicum)"

_plants, 2022, doi:10.3390/plants11212923_

Round 1
Reviewer 1 Report
The present work uses a yeast two-hybrid system to identify proteins interacting with the TOC159 receptor of the plastid outer membrane translocon in tomato. Presented experiments provide new data for researchers studying plastid biogenesis and cellular protein transport. However, similar studies were completed on other plant species, reducing the novelty of the present study. Below you can find a list of my comments.
In the specific Y2H system, prey targeting to ER membrane is sufficient to yield a positive result. That is how positive control used in the study (Ost1) works, as it is not a TOC159 interactor. Is it possible that some identified proteins were not direct TOC159 interactors but were targeted to ER in the yeast cells?
Identified TOC159 interactors are only a fraction of proteins that are imported into plastids, many of which are also expected to interact. The authors discuss possible reasons for a low number of identified interactors. A technical reason could also be considered that the prey library did provide representation only of the most abundant mRNAs. Library validation details could be included in the results. Additionally, Is there any published transcriptomic dataset that would allow comparing how abundant are transcripts of identified interactors as compared to other transcripts? Especially, that hypothesis-based approach allowed the authors to confirm the interaction of TOC159 and TOC34 (Figure 2), which was not detected in the screening experiment.
It could also be considered that TOC159 expressed in yeast cells could have strong interactors in these cells that outcompete some prey library interactors.
Figure 2 does not include the control (i.e., strains with empty prey vector)
Figure 5 In the panels where GFP fluorescence was not detected (i.e., C and H), was the expression of investigated proteins confirmed?
In the results section, bait proteins are described to be preceded by the ER targeting sequence of STE2, but in the methods, it is SUC2 instead. Which is accurate, and why?
Minor points:
Some of the methods-related information could be moved from the results section to the methods section to improve the clarity of the presentation of the results. For example, sentence “7 mg of library DNA plasmid was used for each screen with trans formation efficiency above 2 × 10 clones/mg DNA for all of the library transformations which was sufficient to cover the cDNA libraries multiple times.” is unclear to me, and contains details that do not improve result interpretation. The detailed information (i.e., 7 mg) is combined with vague “multiple times”. It also likely contains an error. Was transformation efficiency 2x10=20 clones per mg of pDNA?
Figure 5 shows the fluorescence of GFP, while the methods section describes it as YFP. Which Is correct?
Figure 5 presents TOC159-2 in addition to TOC159-1, but it is not mentioned in the related part of the methods
The method section mentions “except for p53-large T and hMLH1-BLM” which are not in the results. What were they used for?
I found the description of the plasmids used in the study to be complicated. It would be helpful to present all used plasmid names and their essential details as a table.
The authors declare that the amino acid sequences analyzed in the current study were deposited in the database. However, no specific database entry numbers are given, not allowing verification.
Reviewer 2 Report
In this manuscript, Wang et al., report on the combined use of split-ubiquitin yeast two-hybrid and BiFC analyses to identify proteins that interact with tomato slToc159 isoforms and further studied domain dependence of the isoforms on the specificity of preprotein substrates recognition. While the topic is important and interesting, I am afraid that I find the major portion of the data presented in the manuscript to be unconvincing. The quantity of results included here is considerable, but the dataset as a whole does not reach the required standard in terms of quality. More specific comments are below.
Major points:
1. The prevalent model of preprotein recognition by the TOC complex is that such recognition is mediated via transit-peptide-TOC interaction. However, I find such information (e.g., TP alone or full-length proteins or truncated proteins that include the TP) is missing for the pre-library or identified prey in the Y2H assays. Therefore, the extent to which that the identified interactions reflect preprotein recognition in planta is in doubt. The authors could consider experiments done in Figure 6 Ref. 18. Also, to control for TOC159 interaction specificity with the preprotein (and relevance) the authors could compare interaction with preprotein vs mature protein bates, i.e with or without their N-terminal localization peptide.
2. The BiFC data are puzzling. First, Figure 5A, as it is the full-length slToc159-1 (i.e., receptor at the outer envelope membrane) + RBCS3 (preprotein), localization at the surface of chloroplasts is expected (i.e., outlining chloroplasts) rather than within chloroplasts. It should be possibly to have higher magnification to test this. Second, similarly as in Figure 5E, slToc159-1 A domain + RBCS3 is in the chloroplasts. So, the localization of both the full-length slToc159-1-GFP and slToc159-1 A domain-GFP is required to clarify the localization of slToc159-1 and its A domain.
3. The conclusions in line 388 and line 559 about chloroplast targeting signals in the M domain are not supported by any data shown in the manuscript.
5. It would be helpful if target P and Cell-PLoc predictions (and citations for chloroplast localization, if any) are included in the tables. Some non-photosynthesis proteins, i.e., 60S ribosomal protein (L7a-2), HSP70 and THI026, are cytosolic proteins by name. However, the authors claim that “All screened prey proteins were identified as plastid proteins… (line 215)”.
6. ASS-delta shows significantly different interaction strengths with slToc159-1AGM vs. slToc159-2AGM (Fig. 3A vs. 3C), contradicting the description in Lines 292-293.
7. Lines 326-332. Line 327, what about PQ and UP3? Lines 328-329, ‘two variants of bait’, but listed three baits ‘i.e., slToc159-1GM, slToc159-2AGM, and slToc159-2GM’. Lines 329-330, ‘All four species’, but listed seven proteins. Lines 329-332, It seems these two sentences contradict to each other? i.e., what about the other five? Line 332, should be LacZ.
8. Lines 336-338, the percentages for GS2 and CDP2-1 don’t match Figure 4B.
9. Lines 344-345, the percentages of SAC9, ThiC and FabG don’t match the data shown in Figure 4D.
10. The number of proteins identified is not stated in Results. Did the authors select proteins from a much larger list? In Discussion, the number of proteins identified is stated to be 41. But this is too late in the manuscript to clearly indicate the total number. The number of identified proteins must be stated along with the proportion of those in the tables as soon as they prey are discussed in Results. Related to this, the authors use the term “screened” in reference to the prey proteins. Do they mean “identified”. The screened proteins are the entire set encoded by the 2YH library and tested in the assay for interaction, not just those identified.
Minor points:
1. Define UBPs on first appearance (i.e., line 108).
2. Line 142, should be C-terminal of the cDNA fragments.
3. Fig. 1B, slToc159-2 1162 or 1126 (as in Line 140)?
4. Fig. 1C, labels for lanes 1-4 are required.
5. Line 149, a citation is required for the positive prey control protein pOstI-NubI.
6. Line 184, 1.6 x 10-7? Also line 626.
7. Line 186, should be pPR3-N.
8. Line 191, 2 x 10?
9. Line 305, should be slToc159-2AGM.
10. What is this line 332-333 for?
11. Line 358: …of preprotein … different domains (-A, -G, -M) of slToc159-1…
12. Line 378, citations 66 and 67 are missing. The current reference list stops at 60.
13 Line 415-17 “Since chloroplast development and biogenesis are active in the seedling stage, we generated cDNA libraries using RNAs isolated from early tomato development to ensure the maximum abundance of genes <encoding imported chloroplast proteins (?)> in the library.
14. Line 406, ‘proprotein’ should be ‘preprotein’.
15. Lines 499-501 are repeating line 496-498.
16. Line 525, inputs should by import.
17. Line 526, input should be import.
18. Line 527, ‘A-terminal structure’, A-domain?
19. Line 528, ‘transporter peptides’? transit peptides.
20. Line 25, ‘chloroplast envelope-targeting pathways’ mean the targeting pathways of outer envelope membrane proteins. Here, ‘targeting to chloroplasts’ is more propriate.
21. Lines 172 and 175, are these at or sl proteins?
22. italicize names of species, restriction enzyme.
23. There are excessive methodological details in results eg. line 140-142. 190, elsewhere. For example, should subsection 2.1 be in methods and supplemental data?
24. Fig 1C – show entire WB.
25. Why does Figure 2 shows only Bgal as visual approximation based on color. Figure one quantifies with graph, which is more quantitative and show variance and significance/non-significance.
26. Line 290 “It can be intuitively seen by the PXG method that most photosynthetic prey proteins have a higher interaction strength with slToc159-1 compared to slToc159-2 (Fig. 3).” Analyses should not involve intuition. Comparison of these interactions is hampered by their being on different graphs and lacking results of t-tests.
27. Section 2.6 It is not stated that Pre-proteins were used in these assays. Isn’t the transit peptide required to detect interaction? The terms “Precursor protein” and “proprotein” are used seemly interchangably.
28. Line 391 Chloroplasts and chromoplasts differentiate from proplastids. I don’t know that chloroplasts can transform to chromoplasts. A reference should be provided.
29. Line 443 Yeast does have Hsp70 and Hsp90.
30. Summaries are missing at the end of some of the sections, which makes the manuscript less easy to read than it should be.
Author Response
Detailed Response to Reviewers
Manuscript ID: plants-1907412
Title: Split-Ubiquitin Two-Hybrid Screen for Proteins Interacting with slToc159-1 and slToc159-2,Two Chloroplast Preprotein Import Receptors in tomato(Solanum lycopersicum)
Authors: Qi Wang, Jiang Yue, Chao Zhong Zhang, Jianmin Yan *
Dear Editor: Sammy Han and Dear reviewers:
Thank you for your useful comments on our manuscript. We wish to give a sincere gratitude to referees for reviewing our paper carefully. We apologize for any inconveniences caused by these errors. We have modified the manuscript accordingly, and the response to the referees’ comments are listed point by point below:
Reviewer: 2
Major points:
Comment 1:
The prevalent model of preprotein recognition by the TOC complex is that such recognition is mediated via transit-peptide-TOC interaction. However, I find such information (e.g., TP alone or full-length proteins or truncated proteins that include the TP) is missing for the pre-library or identified prey in the Y2H assays. Therefore, the extent to which that the identified interactions reflect preprotein recognition in planta is in doubt. The authors could consider experiments done in Figure 6 Ref. 18. Also, to control for TOC159 interaction specificity with the preprotein (and relevance) the authors could compare interaction with preprotein vs mature protein bates, i.e with or without their N-terminal localization peptide.
Reply: Thanks for your suggestion. The use of isotopes is strictly regulated in China, so we regret that the experiments performed in Figure 6 Ref 18. were beyond our capabilities. In reference 18, since the authors were unable to generate a functional form of atToc159AG as a bait protein in S. cerevisiae NMY51, a solid-phase binding method was used instead. Several functional bait proteins were able to be produced in our study, so we compare interaction with preprotein vs mature protein (with or without their N-terminal localization peptide) in NMY51, taking TOC159 with RBCS3 as examples. I would like to use Chlorop used in Ref 18. to predict the N-terminal transit peptide sequence information, but since chlorop has been out of service, I used three small RuBisCO small subunit of Arabidopsis (RBCS1B, 2B, 3B) to align with RBCS3 in tomato, and determined the length of the transit peptide of tomato RBCS3, the specificity of slTOC159 with preprotein recognition was determined by NMY51 growth on deficient medium after co-transformation slTOC159 with RBCS3 (with or without cTP), respectively. This result will be presented as a supplementary figure(Fig.S1). I added the relevant content at the end of Section 2.2.
pTSU2-APP and pNubG-Fe65 as positive controls.
pTSU2-APP and pPR3-N as negative controls.
Comment 2:
The BiFC data are puzzling. First, Figure 5A, as it is the full-length slToc159-1 (i.e., receptor at the outer envelope membrane) + RBCS3 (preprotein), localization at the surface of chloroplasts is expected (i.e., outlining chloroplasts) rather than within chloroplasts. It should be possibly to have higher magnification to test this. Second, similarly as in Figure 5E, slToc159-1 A domain + RBCS3 is in the chloroplasts. So, the localization of both the full-length slToc159-1-GFP and slToc159-1 A domain-GFP is required to clarify the localization of slToc159-1 and its A domain.
Reply: Thanks for your suggestion. I totally agree with you. We want to clarify the localization of slToc159-1 and its A domain. We wanted to be able to distinguish more clearly whether slToc159 was localized within the chloroplast or the chloroplast membrane, and we used Arabidopsis protoplasts to further test. However, due to the long fragments of 159-1 and 159-2, which are difficult to be expressed in protoplasts, or because GFP is wrapped by the tertiary spatial structure of slToc159, we failed to observe the full-length localization of Toc159-1 and Toc159-2 after many attempts. We are very sorry for not being able to complete your proposal. We only observed fluorescence signals for GFP fused with the A domain of Toc159-1 and Toc159-2, and the results were unexpected; the A domain was localized in the cytoplasm, indicating that the a structural region did not contain any membrane targeting signal. However, the results of BiFC show that the A domain can indeed recognize the precursor protein, indicating the importance of the A domain. Here we focus on A domain, and further tests show that it does not contain membrane targeting signals. Further exploration of membrane targeting signals seems to be beyond our research scope. I deleted the description of domain A containing membrane targeted signals in the manuscript and modified it. You can see the localization results of A domain in Arabidopsis protoplasts in Figure 7
Comment 3:
The conclusions in line 388 and line 559 about chloroplast targeting signals in the M domain are not supported by any data shown in the manuscript.
Reply: Thanks for your comment. I totally agree with you, I have deleted the relevant conclusions in the manuscript.
Comment 4:
It would be helpful if target P and Cell-PLoc predictions (and citations for chloroplast localization, if any) are included in the tables. Some non-photosynthesis proteins, i.e., 60S ribosomal protein (L7a-2), HSP70 and THI026, are cytosolic proteins by name. However, the authors claim that “All screened prey proteins were identified as plastid proteins… (line 215)”.
Reply: Thanks for your useful suggestion. I'm very sorry for the mistakes in the manuscript. I have included the site-predicted subcellular localization results in Tables 1 and 2. Taking into account your suggestion, it is necessary to predict the localization of all non-photosynthetic proteins again using UniProt database and Plant-PLoc, and finally we identified 60S ribosomal protein (L7a-2), HSP70 and THI026 as cytoplasmic localization. In addition, Calcium-binding protein (KIC) and Serine hydroxymethyl transferase (SHT) were also re-predicted as mitochondrial-localized proteins, I have revised the relevant parts of the manuscript.
Comment 5:
ASS-delta shows significantly different interaction strengths with slToc159-1AGM vs. slToc159-2AGM (Fig. 3A vs. 3C), contradicting the description in Lines 292-293.
Reply: Thanks for your comment. I'm very sorry for such a mistake. I rechecked the results in the picture and the corresponding content, and made modifications, marked in red font.
Comment 6:
Lines 326-332. Line 327, what about PQ and UP3? Lines 328-329, ‘two variants of bait’, but listed three baits ‘i.e., slToc159-1GM, slToc159-2AGM, and slToc159-2GM’. Lines 329-330, ‘All four species’, but listed seven proteins. Lines 329-332, It seems these two sentences contradict to each other? i.e., what about the other five? Line 332 should be LacZ.
Reply: Thanks for your comment. Here I mean that of the 15 prey proteins identified using 159-2AGM alone as bait, 7 were observed to interact with the other three bait variants. While both UP3 and PQ could interact with the three variants, UP3 was identified in both screening assays using 159-1AGM and 159-2AGM as bait, and PQ was only identified in the screening assay using 159-1AGM as bait identified and therefore not described here. Lines 328-329, here are indeed three variants of the bait, I have revised and marked in red font. Lines 329-330, I have changed to seven proteins.
Lines 329-332, there seems to be a contradiction here, I have made changes in the manuscript and marked in red font.
Comment 7:
Lines 336-338, the percentages for GS2 and CDP2-1 don’t match Figure 4B.
Reply: Thanks for your comment. The numbers in brackets represent the relative increase in the interaction strength of slToc159-1AGM compared to slToc159-1GM. My calculation formula is (slToc159-1AGM-slToc159-1GM) / slToc159-1GM= relative increase. I calculated and confirmed the values in the picture again and the data is correct. I would like to ask if this calculation method is correct and look forward to your reply.
Comment 8:
Lines 344-345, the percentages of SAC9, ThiC and FabG don’t match the data shown in Figure 4D.
Reply: Thanks for your comment. The numbers in brackets represent the relative increase in the interaction strength of slToc159-1AGM compared to slToc159-1GM. My calculation formula is (slToc159-2AGM-slToc159-2GM) / slToc159-2GM= relative increase. I calculated and confirmed the values in the picture again and the data is correct. I would like to ask if this calculation method is correct and look forward to your reply.
Comment 9:
The number of proteins identified is not stated in Results. Did the authors select proteins from a much larger list? In Discussion, the number of proteins identified is stated to be 41. But this is too late in the manuscript to clearly indicate the total number. The number of identified proteins must be stated along with the proportion of those in the tables as soon as they prey are discussed in Results. Related to this, the authors use the term “screened” in reference to the prey proteins. Do they mean “identified”. The screened proteins are the entire set encoded by the Y2H library and tested in the assay for interaction, not just those identified.
Reply: Thanks for your comment and suggestion. I totally agree with you. I describe the number of proteins identified is 41 in the results section 2.2 and marked in red font. I did not select proteins from a much larger list. The 41 proteins identified are all described in the table. All subsequent experiments were based on the identified 41 proteins listed in Tables 1 and 2. The term “screened” do mean “identified”. Your suggestion is very correct, the use of the term “screened” does cause some misunderstandings in the manuscript. I have changed “screened” to “identified” in the manuscript and marked in red font.
Minor points:
Comment 1:
Define UBPs on first appearance (i.e., line 108).
Reply: Thanks for your comment. ubiquitin-specific proteases (UBPs). I have revised in the manuscript and marked in red font
Comment 2:
Line 142 should be C-terminal of the cDNA fragments.
Reply: Thanks for your comment. I'm very sorry for this mistake. I have made revisions in the manuscript and marked in red font.
Comment 3:
Fig. 1B, slToc159-2 1162 or 1126 (as in Line 140)?
Reply: Thanks for your comment. I'm very sorry for this mistake. I have made revisions in the manuscript and marked in red font.
Comment 4:
Fig. 1C, labels for lanes 1-4 are required.
Reply: Thanks for your suggestion. I have added labels for lanes 1-4 in Fig. 1C.
Comment 5:
Line 149, a citation is required for the positive prey control protein pOstI-NubI.
Reply: Thanks for your suggestion. I have added a citation for the positive prey control protein pOstI-NubI in the manuscript.
Comment 6:
Line 184, 1.6 x 10-7? Also line 626.
Reply: I am very sorry for this mistake. Line 184, Correct is 1.6 x 107. line 626, Correct is 4.0 x 106. I have made revisions in the manuscript and marked in red font.
Comment 7:
Line 186, should be pPR3-N.
Reply: Thanks for your comment. I have made revisions in the manuscript and marked in red font.
Comment 8:
Line 191, 2 x 10?
Reply: I am very sorry for this mistake. Line 191, Correct is 2 x 106.
Comment 9:
Line 305 should be slToc159-2AGM.
Reply: I am very sorry for this mistake. I have made revisions in the manuscript and marked in red font.
Comment 10:
What is this line 332-333 for?
Reply: Thanks for your comment. What I want to express here is that the two prey proteins SLP1 and SHT can only interact with slToc159-2AGM, the way I express it here is really strange and puzzling, I have made revisions in the manuscript and marked in red font.
Comment 11:
Line 358: …of preprotein … different domains (-A, -G, -M) of slToc159-1…
Reply: Thanks for your comment. I'm very sorry for such a mistake. I have made revisions in the manuscript and marked in red font.
Comment 12:
Line 378, citations 66 and 67 are missing. The current reference list stops at 60.
Reply: Thanks for your comment. I am very sorry for that. There should be none here, I have deleted it
Comment 13:
Line 415-17 “Since chloroplast development and biogenesis are active in the seedling stage, we generated cDNA libraries using RNAs isolated from early tomato development to ensure the maximum abundance of genes <encoding imported chloroplast proteins (?)> in the library.
Reply: Thanks for your comment and suggestion. I totally agree with you, your description makes the manuscript easier to understand, I have revised the manuscript as you described. RNAs isolated from early tomato development to ensure the maximum abundance of genes encoding imported chloroplast proteins in the library.
Comment 14:
Line 406, ‘proprotein’ should be ‘preprotein’.
Reply: Thanks for your comment. I'm very sorry for such a mistake. I have made revisions in the manuscript.
Comment 15:
Lines 499-501 are repeating line 496-498.
Reply: Thanks for your comment. I have removed duplicates.
Comment 16:
Line 525, inputs should by import.
Reply: Thanks for your suggestion. I have made revisions in the manuscript and marked in red font.
Comment 17:
Line 526, inputs should by import.
Reply: Thanks for your suggestion. I have made revisions in the manuscript and marked in red font.
Comment 18:
Line 527, ‘A-terminal structure’, A-domain?
Reply: Thanks for your comment. I'm very sorry for that. 'A-terminal structure' does mean A-domain. In order to keep the description consistent and not cause misunderstanding, I have changed it.
Comment 19:
Line 528, ‘transporter peptides’? transit peptides.
Reply: Thanks for your comment. I have made revisions in the manuscript and marked in red font.
Comment 20:
Line 25, ‘chloroplast envelope-targeting pathways’ mean the targeting pathways of outer envelope membrane proteins. Here, ‘targeting to chloroplasts’ is more propriate.
Reply: Thanks for your suggestion. I totally agree with your suggestion and thank you very much. I have made revisions in the manuscript and marked in red font.
Comment 21:
Lines 172 and 175, are these at or sl proteins?
Reply: Thanks for your comment. I'm sure it's sl protein here, I'm so sorry for such a mistake. I have made revisions in the manuscript and marked in red font.
Comment 22:
italicize names of species, restriction enzyme.
Reply: Line 488, names of species;line140, 171, 633, restriction enzyme. I have italicized them and marked in red font.
Comment 23:
There are excessive methodological details in results eg. line 140-142. 190, elsewhere. For example, should subsection 2.1 be in methods and supplemental data?
Reply: Thanks for your suggestion. Excessive methodological details on line 140-142 I have moved to the methods section. Considering your suggestion, i have also moved the excessively detailed description in 2.2 to the Methods section, e.g., 7 mg of library DNA plasmid was used for each screen.
Comment 24:
Fig 1C – show entire WB.
Reply: Thanks for your suggestion. At the time of manuscript submission, I provided the entire WB, however showing the entire WB in Figure 1 would make Figure 1 difficult to typeset into a regular figure, so I would like to ask if it is possible to display the entire WB as a supplementary file.
Comment 25:
Why does Figure 2 shows only B-gal as visual approximation based on color. Figure one quantifies with graph, which is more quantitative and show variance and significance/non-significance.
Reply: Thanks for your comment. In Figure 2, we just want to ask whether slToc159 and slToc34 have potential interaction ability, because they have not been identified in the library, I just want to prove the interaction between them through the activation of lacZ gene, not for comparison the strength of their interaction, therefore no further quantitative determination of β-gal activity was performed. I deleted the last sentence in section 2.3, because here I just wanted to show that they interact, and it is unreliable for me to describe the strength of their interaction by visual observation
Comment 26:
Line 290 “It can be intuitively seen by the PXG method that most photosynthetic prey proteins have a higher interaction strength with slToc159-1 compared to slToc159-2 (Fig. 3).” Analyses should not involve intuition. Comparison of these interactions is hampered by their being on different graphs and lacking results of t-tests.
Reply: Thanks for your comment. I agree with you so much that I have changed the relevant expressions in the manuscript.
Most photosynthetic prey proteins appear to interact more strongly with slToc159-1 than with slToc159-2, as observed by the PXG method and subsequently demonstrated by quantification of β-gal activity.
Comment 27:
Section 2.6 It is not stated that Pre-proteins were used in these assays. Isn’t the transit peptide required to detect interaction? The terms “Precursor protein” and “proprotein” are used seemly interchangably.
Reply: Thanks for your comment. I fully agree with you. The transit peptides are essential for the interaction to occur. I did not state that RBCS3 is a precursor protein, which is not rigorous, so I added a description after RBCS3 in section 2.6, including 54 N-terminal amino acid cTP.
Comment 28:
Line 391 Chloroplasts and chromoplasts differentiate from proplastids. I don’t know that chloroplasts can transform to chromoplasts. A reference should be provided.
Reply: Thanks for your comment. I added three relevant references.
- Egea I. et al. Chromoplast differentiation: current status and perspectives. Plant Cell Physiol. 2010;51(10):1601-11.
- Dhami N. Chloroplast-to-chromoplast transition envisions provitamin A biofortification in green vegetables. Plant Cell Rep. 2021;40(5):799-804.
- Ling, Q. et al. The chloroplast-associated protein degradation pathway controls chromoplast development and fruit ripening in tomato. Nat Plants 7, 655–666 (2021).
Comment 29:
Line 443 Yeast does have Hsp70 and Hsp90.
Reply: Thanks for your comment. I deleted HSP70 and HSP90 and added AKR2 (AnKyrin Repeat-containing protein 2) described as a plant-specific chaperone.
Comment 30:
Summaries are missing at the end of some of the sections, which makes the manuscript less easy to read than it should be.
Reply: Thanks for your comment. I added a summary at the end of 2.4 and 2.5

Round 2
Reviewer 1 Report
The revised version of the manuscript is substantially improved and includes additional data. I appreciate the explanations provided by the authors. I support the publication of the study.
To avoid confusion, the authors can consider adding the interaction between bait and prey proteins to the Figure 1A graphics. In the currently presented model, only Cub and NubG fragments appear to interact.
Reviewer 2 Report
1. The results in Fig S1 address this point sufficiently. The text for Fig S1 should refer to the names of constructs and the cytoplasmic protein prey controls. The legend should provide a more complete description of the experiments.
2. It is possible get higher magnification with sufficient resolution to distinguish the chloroplast interior and boundary on any fluorescence microscope. It is not necessary to go to protoplasts. see Fig1 in https://doi.org/10.1073/pnas.172390399
